# 2-(4-Benzyloxy-3-methoxyphenyl)-5-(carbethoxyethylene)-7-methoxy-benzofuran, a Benzofuran Derivative, Suppresses Metastasis Effects in P53-Mutant Hepatocellular Carcinoma Cells

**DOI:** 10.3390/biomedicines11072027

**Published:** 2023-07-19

**Authors:** Tsui-Hwa Tseng, Yi-Chia Shao, Yean-Jang Lee, Huei-Jane Lee

**Affiliations:** 1Department of Medical Applied Chemistry, Chung Shan Medical University, Taichung 40201, Taiwan; tht@csmu.edu.tw (T.-H.T.); a14253681@gmail.com (Y.-C.S.); 2Department of Medical Education, Chung Shan Medical University Hospital, Taichung 40201, Taiwan; 3Department of Chemistry, National Changhua University of Education, Changhua 50007, Taiwan; leeyj@cc.ncue.edu.tw; 4Department of Biochemistry, School of Medicine, College of Medicine, Chung Shan Medical University, Taichung 40201, Taiwan; 5Department of Clinical Laboratory, Chung Shan Medical University Hospital, Taichung 40201, Taiwan

**Keywords:** benzofuran, ailanthoidol, p53, hepatocellular carcinoma, epithelial–mesenchymal transition, metastasis

## Abstract

2-(4-Benzyloxy-3-methoxyphenyl)-5-(carbethoxyethylene)-7-methoxy-benzofuran (BMBF), a benzofuran derivative, is an intermediate found in the process of total synthesis of ailanthoidol. Benzofuran derivatives are a class of compounds that possess various biological and pharmacological activities. The present study explored the anti-metastasis effects of BMBF in hepatocellular carcinoma (HCC). Our preliminary findings indicate that BMBF suppresses the proliferation and changes the morphology of Huh7—an HCC cell line with a mutated p53 gene (Y220C). According to a scratching motility assay, non-cytotoxic concentrations of BMBF significantly inhibited the motility and migration in Huh7 cells. BMBF upregulated the expression of E-cadherin and downregulated the expression of vimentin, Slug, and MMP9, which are associated with epithelial–mesenchymal transition (EMT) and metastasis in Huh7 cells. BMBF decreased the expression of integrin α7, deactivated its downstream signal FAK/AKT, and inhibited p53 protein levels. Cell transfection with p53 siRNA resulted in the prevention of cell invasion because of the reduction in integrin α7, Slug, and MMP-9 in Huh7 cells. BMBF had anti-metastatic effects in PLC/PRF/5—an HCC cell line with R249S, a mutated p53 gene. Our findings indicate that BMBF has anti-metastatic effects in downregulating p53 and mediating the suppression of integrin α7, EMT, and MMP-9 in HCC cells with a mutated p53 gene.

## 1. Introduction

Hepatocellular carcinoma (HCC) is one of the most common tumors worldwide. Approximately 75–85% of liver cancer mortality is the result of HCC [1]. Early identification of HCC is difficult; therefore, most cases of HCC are discovered at a late stage [2]. Early-stage HCC is generally treated with a combination of surgery, radiotherapy, and chemotherapy [3]. A modern, effective treatment for HCC is liver transplantation; however, organ shortages, perioperative risk, and the strict requirements for appropriate pairing limit the accessibility of liver transplantation. Despite the progress that has been made in therapeutic approaches, HCC recurrence and metastasis rates remain high, leading to unfavorable prognoses [4,5]. The development of agents to prevent HCC metastasis is one strategy to increase the survival rate of patients with HCC.

Wild-type p53 protein (WTP53) plays a key role in cell apoptosis to regulate the cell cycle after DNA damage [6]. Cells with a mutated p53 gene may evade apoptosis after DNA damage and potentially become cancerous. Mutations in the p53 gene are the most common type of gene change in HCC, with an average mutation frequency of 30% [6]. Cells with a mutated p53 gene lose their tumor-suppressing function and promote tumorigenesis and metastasis [7]. In addition, WTP53 and mutated p53 protein (MTP53) are involved in the regulation of cell migration and invasion in cancer cell metastasis [8]. Wang et al. reported that knockdown of WTP53 enhances epithelial-to-mesenchymal transition (EMT), migration, and metastasis in liver cancer cells [9]. In an esophageal cell model, Ohashi et al. observed that MTP53 (R175H) cooperates with the epithelial growth factor receptor to promote the EMT phenotype upon treatment with TGFβ [10]. Lenfer et al. indicated that MTP53 enhanced metastasis to aggravate the tumor phenotype in a bitransgenic tumor model [11]. WTP53 and MTP53 have tumor-suppressive and oncogenic roles, respectively. MTP53 promotes EMT, whereas WTP53 prevents EMT [12]. Therapies that decrease MTP53 expression or target MTP53 may have potential as a means of preventing HCC metastasis.

Benzofuran derivatives are a class of compounds found in higher plants that have attracted the attention of chemists and pharmacologists because of their various biological and pharmacological activities, which include anti-inflammatory, antimicrobial, antiviral, anti-hyperglycemic, and antitumor activities. In addition to isolating benzofuran derivatives from natural products, medicinal chemists are investigating methods of synthesizing benzofuran rings for application in drugs [13,14]. The benzofuran derivative 2-(4-benzyloxy-3-methoxyphenyl)-5-(carbethoxyethylene)-7-methoxy-benzofuran (BMBF) is an intermediate produced in the process of total synthesis of ailanthoidol—a neolignan originating from the bark of *Zanthoxylum ailanthoidol* (Rutaceae) [15]. Ailanthoidol had an antitumor effect in a multistep mouse model of skin cancer [16]. Ailanthoidol suppressed TGF-β1-promoted migration and invasion in HepG2 cells [17] and inhibited the proliferation of Huh7 cells [18]. Although ailanthoidol exhibits antitumor potential, the biological mechanism of BMBF remains unclear. Our preliminary study showed that BMBF cannot alter the cell viability in HepG2 hepatoblastoma cells (Appendix A) but suppresses the proliferation and morphological changes in Huh7 hepatocellular carcinoma cells. Because HepG2 expresses wild-type p53, whereas Huh7 has a mutated p53 gene, we presumed that BMBF perhaps exhibits a pharmaceutical effect in the HCC cells with p53 mutation. Therefore, the present study investigated the anti-metastatic and modulatory effects of BMBF in HCC cells with a mutated p53 gene.

## 2. Materials and Methods

### 2.1. Materials

Dulbecco’s modified Eagle’s medium (DMEM), minimum essential medium (MEM), phosphate-buffered saline (PBS), fetal bovine serum (FBS), penicillin, streptomycin, and trypsin-EDTA were purchased from Gibco Ltd. (Grand Island, NY, USA). Primary antibodies against integrin α7, Slug, E-cadherin, vimentin, MMP9, p53 (DO-1), GADPH, and actin were obtained from Santa Cruz Biotechnology, Inc., CA, USA. Matrigel was obtained from Collaborative Research (Bedford, MA, USA). TRITC-conjugated phalloidin, β-actin antibody, and other chemicals were purchased from Sigma-Aldrich (St. Louis, MO, USA). BMBF, shown in Figure 1, was provided by Dr. Yean-Jang Lee and synthesized from 5-bromo-2-hydroxy-3-methoxybenzaldehyde, as previously reported [15]. Anti-FAK, anti-p-FAK, anti-AKT, and anti-p-AKT were purchased from Cell Signaling Technology (Beverly, MA, USA).

### 2.2. Cell Culture and Cell Viability Assay

The human liver cancer cell line Huh7 (p53 mutant in Y220C) was obtained from the Food Industry Research and Development Institute (Hsinchu, Taiwan) and cultured in DMEM supplemented with 10% FBS, 1% penicillin/streptomycin, 1% essential amino acids, and 1 mM glutamine. PLC/PRF/5 (p53 mutant in R249S) cells were cultured in MEM supplemented with 10% FBS and 1% penicillin/streptomycin. The cells were maintained at 37 °C in a humidified atmosphere with 5% CO_2_. To evaluate the cytotoxicity of BMBF in HCC cells, Cell Counting Kit-8 (CCK-8, Sigma-Aldrich, St. Louis, MO, USA) was used. Briefly, 3 × 10^3^ cells were seeded onto a 96-well Petri dish and then treated with various concentrations of BMBF for the indicated duration. Subsequently, 10 μL of CCK-8 solution was added to incubate with the medium for 3 h; the absorbance was read at a wavelength of 450 nm using an ELISA reader (SpectraMax M5, Molecular Devices, Downingtown, PA, USA).

### 2.3. Microscopic Examination

After treatment under the indicated conditions, Huh7 cells were fixed with 4% paraformaldehyde for 10 min and permeabilized with 0.1% triton X-100 in PBS for 5 min. The cell morphology was assessed by phase-contrast microscopy. In addition, cytoskeletal changes (F-actin) were analyzed through fluorescence microscopy by staining with TRITC-conjugated phalloidin (500 ng/mL) for 1 h. Images were acquired using a fluorescence microscope (Nikon Microscope SE, Nippon Kogaku KK, Tokyo, Japan) at 400× or 200× magnification.

### 2.4. Scratch Motility Assay

Huh7 cells (2.5 × 10^5^ cells/mL) were seeded onto a 6-well plate and grown overnight to confluence. The monolayer was scratched with a yellow pipette tip, washed with PBS to remove floating cells, and photographed (0 h) before being treated with BMBF (0–5 μM). After being photographed (24 h), the cells that were motile in the scratched area were counted in 5 randomly selected fields (100× magnification) by digital planimetry using ImageJ software. The area of cell migration was expressed as a percentage of the initial area (0 h). Data are represented as the mean ± SD of three independent experiments.

### 2.5. Cell Migration and Invasion Assay

Cell migration and invasion assays were performed using a Boyden chemotaxis chamber. The upper culture chamber consisted of a polycarbonate filter (pore size, 8 μm) coated with (for invasion) or without (for migration) a uniform layer of 40 μg/cm^2^ of Matrigel basement membrane matrix in the upper compartment of the chemotaxis chamber. Huh7 cells were pretreated with BMBF (0–5 μM) for 24 h. The cells were harvested, and 6 × 10^4^ cells/well were suspended in serum-free media and then placed in the upper chamber. The complete growth medium with 10% FBS was placed in the lower chamber. After incubation for 24 h, the cells on the upper surface of the filter were wiped with a cotton swab. The cells on the lower surface of the filter were fixed for 10 min with methanol and stained with Giemsa for 1 h, and the cells that had migrated or invaded into the lower surface of the filter were sequentially counted by light microscopy (200×). The experiment was performed in triplicate; in each filter, the cells from 5 randomly selected fields were counted to represent the data as the mean ± SD.

### 2.6. Preparation of Total Cell Extracts and Immunoblot Analysis

Cells were plated onto 10 cm^2^ dishes at a density of 1 × 10^6^ cells/mL and treated with BMBF for 24 h. To prepare the whole-cell extracts, the cells were harvested and suspended in a lysis buffer (50 mM Tris, 5 mM EDTA, 150 mM NaCl, 1% NP40, 0.5% deoxycholic acid, 1 mM sodium orthovanadate, 81 μg/mL aprotinin, 170 mg/mL leupeptin, 100 μg/mL PMSF; pH 7.5). After reacting for 30 min at 4 °C, the mixtures were centrifuged at 10,000× *g* for 10 min, and the supernatants were collected as the whole-cell extracts. The protein content was determined by the Bradford protein assay (Kenlor Industries, Costa Mesa, CA, USA). Equal amounts of protein sample were subjected to 8–12% SDS–polyacrylamide gel electrophoresis to separate and then electrotransferred to nitrocellulose membranes (Sartorius Co., Goettingen Germany). They subsequently reacted with the primary antibodies (i.e., anti-E-cadherin, anti-vimentin, anti-Slug, anti-MMP-9, and anti-integrin α7). Anti-GADPH or anti-β-actin was used as the internal control. The secondary antibody was a peroxidase-conjugated goat anti-mouse or anti-rabbit antibody. After completing the procedures, the bands were exposed by enhanced chemiluminescence using a commercial enhanced chemiluminescence (ECL) kit (Immobilon^TM^ Western, Millipore Co., Billerica, MA, USA).

### 2.7. Transfection of p53siRNA

Next, 3 × 10^3^ Huh7 cells were seeded in 96-well dishes, or 4 × 10^5^ cells on 10 cm dishes. Following incubation overnight, p53 siRNA (40 nM and 80 nM) or control siRNA (40 nM) (Santa Cruz Biotechnology, Santa Cruz, CA, USA) was transfected using T-Pro NTR II transfection reagent according to the manufacturer’s instructions (T-Pro Biotechnology Co., New Taipei City, Taiwan). The p53 siRNAs (sense: 5′-AGA-CCU-AUG-GAA-ACU-ACU-Utt-3′) were purchased from GeneDireX Inc. (Taoyuan City, Taiwan). Following incubation for 48 h, the cells were treated with or without BMBF for 24 h, and then the viable cells were added to the upper chamber of the Boyden chamber for invasion assay, or the total cell lysate was prepared for immunoblotting analysis.

### 2.8. Statistical Analysis

Statistical significance was determined by one-way analysis of variance with the post hoc Dunnett’s test. *p*-values lower than 0.05 were considered statistically significant.

## 3. Results

### 3.1. BMBF Suppressed the Viability of Huh7 Cells

The cytotoxicity of BMBF in Huh7 cells was assessed using the CCK-8 assay. Huh7 cells were treated with various concentrations of BMBF (0, 5, 10, 20, 40, and 80 μM) for 24 and 48 h. Treatment with concentrations of BMBF greater than 5 μM for 24 and 48 h significantly suppressed the viability of the Huh7 cells (Figure 2). In the Huh7 cells, the IC_50_ value of BMBF at 24 h was 48.22 μM, and at 48 h it was 38.15 μM. The various concentrations of BMBF used in this study exhibited no cytotoxicity in normal hepatocytes (Appendix A).

### 3.2. BMBF Reduced the Cytoskeletal Changes and Inhibited Motility, Migration, and Invasion in Huh7 Cells

Metastasis occurs in the majority of cancer deaths and is a complex process consisting of tumor cell motility away from the primary site, migration into the vasculature, invasion into surrounding parenchyma, and growth at the metastatic sites. A crucial element of this process is the remodeling of the cytoskeleton [19]. To evaluate the anti-metastatic potential of BMBF, non-cytotoxic concentrations of BMFB were used in this study. First, we investigated the effect of BMBF on the actin cytoskeleton of Huh7 cells by using TRITC-conjugated phalloidin. In our preliminary observation, Huh7 cells—cells from an aggressive HCC cell line—exhibited lamellipodium protrusion and a more intensely stained F-actin cytoskeleton. Nonetheless, the F-actin cytoskeleton was reduced when treated with BMBF (Figure 3A). Cytoskeletal alterations are associated with cell motility; therefore, we investigated the effect of BMBF on the motility of Huh7 cells by using a scratch motility assay. BMBF dose-dependently inhibited wound closure (Figure 3B,C). Furthermore, the Boyden chamber assay revealed that BMBF at a concentration of 1, 2.5, or 5 μM significantly suppressed Huh7 cells’ migration and invasion (Figure 4A,B). These findings indicate that BMBF has in vitro anti-metastatic potential in HCC cells.

### 3.3. Inhibitory Effects of BMBF on EMT-Related Proteins and Integrin α7 in Huh7 Cells

EMT is a biological process in which polarized epithelial cells undergo multiple internal biological changes and transition into a mesenchymal phenotype; the process is highly mobile and invasive [20]. EMT plays an essential role in the progression and metastasis of HCC [21]. EMT involves the loss of E-cadherin and the production of vimentin, which enables cells to migrate and invade surrounding tissue. Matrix metalloproteinases (MMPs) are also involved in this process [22]. We investigated the effects of BMBF on the levels of E-cadherin, vimentin, MMP-9, and Slug, which is the transcription factor involved in EMT-related protein expression in Huh7 cells. BMBF upregulated the expression of E-cadherin and suppressed vimentin, Slug, and MMP-9 (Figure 5A). Integrins are membrane protein receptors that trigger distinct signaling and play a key role in the propagation and progression of cancer [23]. Integrin α7 expression was reported to be higher in metastatic HCC cells than in non-metastatic cells [24], and integrin α7 was reported to be overexpressed in Huh7 cells [25]. We analyzed the effects of BMBF on the expression of integrin α7 and the phosphorylation of its downstream signal mediators, such as FAK and AKT [23]. BMBF suppressed the expression of integrin α7 and decreased the phosphorylation of FAK and AKT (Figure 5B).

### 3.4. BMBF Suppressed the Invasion in Huh7 Cells with p53 Knockdown

EMT in HCC cells involves p53 [9]. We investigated the effect of BMBF on p53 expression in Huh7 cells. BMBF decreased p53 expression in the Huh7 cells (Figure 6A). The invasion ability of Huh7 cells transfected with p53 siRNA was assessed by using the Boyden chamber assay; p53 siRNA significantly inhibited invasion and suppressed the expression of integrin α7, Slug, and MMP9 in Huh7 cells. BMBF-induced downregulation of p53 has anti-metastatic potential.

### 3.5. Anti-Invasion of BMBF in PLC/PRF/5 cells

We evaluated the anti-invasion potential of BMBF in PLC/PRF/5 hepatocellular carcinoma cells with the p53 mutant R249S by using the Boyden chamber assay. BMBF significantly inhibited the invasion effect (Figure 7A,B). Consistent with the effect of BMBF in the Huh7 cells, BMBF suppressed the expression of p53, integrin α7, and MMP9. In addition, BMBF upregulated E-cadherin and downregulated vimentin and the EMT-related transcription factor Slug (Figure 7C).

## 4. Discussion

In the present study, the benzofuran derivative BMBF suppressed migration and invasion in HCC cells with a mutated p53 gene. The underlying mechanisms involve the upregulation of E-cadherin and the downregulation of vimentin, Slug, and MMP-9. BMBF decreased integrin α7 expression, deactivated FAK/AKT, and inhibited the expression of p53 to suppress metastasis (Figure 8). Our findings indicate that BMBF is a potential anti-metastatic agent in HCC cells with a p53 mutation.

Metastasis is the main reason for the failure of cancer therapy. EMT is an essential process in cancer metastasis. EMT allows normal hepatic epithelial cells to undergo multiple biological changes that enable them to assume a mesenchymal phenotype, which enhances the cells’ migration and invasion capacity and increases their resistance to apoptosis [21]. Aberrant activation of EMT is crucial in cancer metastasis and involves multiple molecular mechanisms and signal transduction pathways, the hallmark of which is the downregulation of E-cadherin and upregulation of vimentin. The transcription factors Slug and Twist induce EMT [26]. MTP53 promotes the expression of several EMT-related transcription factors [9]. In the present study, BMBF reduced the expression of p53 in Huh7 and PLC/PRF/5 cells with mutated p53 genes. Furthermore, BMBF upregulated the expression of E-cadherin and downregulated the expression of vimentin and the EMT-associated transcription factor Slug. MTP53 protein levels may be affected by miRNA or enzyme-controlled stability [27]. Our results indicate that BMBF has anti-metastatic properties in HCC cells; however, its underlying mechanisms in reducing MTP53 warrant further investigation.

Integrins are transmembrane receptors built up by the αβ-heterodimer. Several integrins are downregulated in tumor tissues [23]. Integrin α7 is a key regulator in tumor propagation and has cancer stem cell properties [28,29]. Integrin α7 expression is high in various cancer cells, including mesothelioma and Huh7 cells [25,30]. Wu et al. reported that integrin α7 knockdown suppressed HCC progression and inhibited EMT in HCCs [25]. Hass et al. observed that integrin α7 regulates several signal pathways, including the FAK/AKT pathway, promoting cell proliferation and metastasis [31]. Moreover, integrin α7 is associated with negative clinical outcomes in patients with HCC and regulates cancer stem cell markers [32]. In the present study, BMBF reduced the integrin α7 levels and deactivated the downstream FAK/AKT signaling pathway. This demonstrates that BMBF-induced downregulation of integrin α7 prevents HCC metastasis. Whether BMBF regulates cancer stem cell markers in HCC requires further elucidation.

Deletion or mutation of p53 occurs in approximately 50% of patients with cancer and results in the loss of its tumor-suppression function. Accumulating evidence indicates that mutation of p53 leads to oncogenic gain-of-function effects, such as promoting cancer metastasis [8]. Cancer metastasis contributes to over 90% of cancer-associated deaths [33]. Metastasis involves a sequence of events from cancer cell invasion at the primary tumor site to outgrowth of metastatic colonies at distant organs. In order to survive during this multistep metastasis cascade, tumor cells reprogram gene expression, rewind metabolisms, and regulate intracellular and intercellular signaling. MTP53 is an important regulator of metastasis. MTP53 induces expression of Slug, which is a transcription factor of EMT, in HCT116 colon carcinoma cells [34]. In addition to regulating transcription factors, the crosstalk between MTP53 and TGFβ signaling is involved in the regulation of cell motility and invasion [35,36]. MTP53 may also enhance integrins’ expression and/or regulate their N-glycosylation to contribute to cancer cell–ECM interaction and metastasis [37,38]. MTP53 is suggested to induce several receptor tyrosine kinase pathways to promote tumor invasion [39,40]. Moreover, tumor cells with MTP53 generate a pro-invasion niche through releasing exosomes, which can be taken up by neighboring tumor-associated macrophages to create a supportive microenvironment and drive tumor progression to a more aggressive state [41,42]. Mutation of p53 leads to oncogenic gain-of-function properties and results in cancer metastasis [8]. In the present study, non-cytotoxic concentrations of BMBF exhibited anti-migration and anti-invasion effects, along with downregulating MTP53 levels in HCC cells. Mutations of R249S in p53, which represent a gain of function, are phosphorylated by CDK4/cyclin D1 and then translocated into the nucleus. In the nucleus, R249S binds to and augments c-Myc activity, resulting in an increase in ribosome biogenesis and proliferation [43]. Y220C mutations in p53 can cause the dedifferentiation of hepatocytes in response to oncogenic stimuli, which may result in the growth of malignant reprogrammed progenitor cells [44].

Benzofuran belongs to a critical class of heterocyclic compounds or fragments that are present in many drugs [10]. Due to the biological and medicinal importance of benzofuran, benzofuran derivatives have attracted the attention of scientists [13,45]. Ailanthoidol, a natural benzofuran, has exhibited antitumor potential [16,17,18]. Ailanthoidol, through downregulation of MTP53 and deactivation of the STAT3 pathway, has an antiproliferative effect in Huh7 cells. Benzofuran derivatives, through HIF-1 inhibition, also have an antiproliferative effect, especially against p53-independent (or p53-deleted) malignant tumors [32]. In addition to antitumor and antiproliferative effects, the present study revealed that benzofuran derivatives can suppress tumor metastasis in HCC cells with MTP53. Although BMBF is an intermediate from the process of total synthesis of ailanthoidol, it has anti-metastatic bioactivity. Although the mechanism of BMBF in decelerating cancer progression is different from that of ailanthoidol, the present study indicates that the intermediates with similar structures produced in the process of chemical synthesis may also possess bioactivity. This implies that the use of synthetic intermediates can expand the application of drug synthesis.

## 5. Conclusions

Our findings indicate that mutations in p53 affect tumor growth and the regulation of metastasis. In vivo studies investigating physiological responses should be conducted to verify the effects and mechanisms of BMBF. Whether BMBF inhibits the metastasis of p53-independent (p53 deleted) malignant tumors requires further clarification.

## Figures and Tables

**Figure 1 biomedicines-11-02027-f001:**
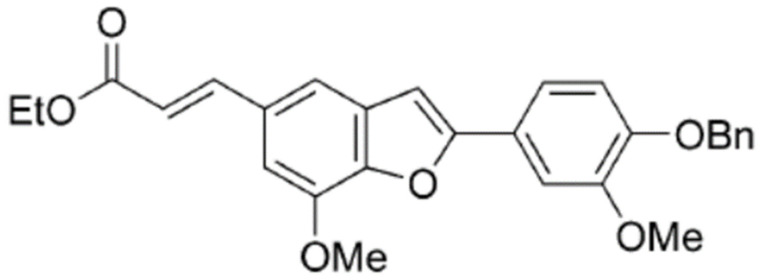
Chemical structure of BMBF.

**Figure 2 biomedicines-11-02027-f002:**
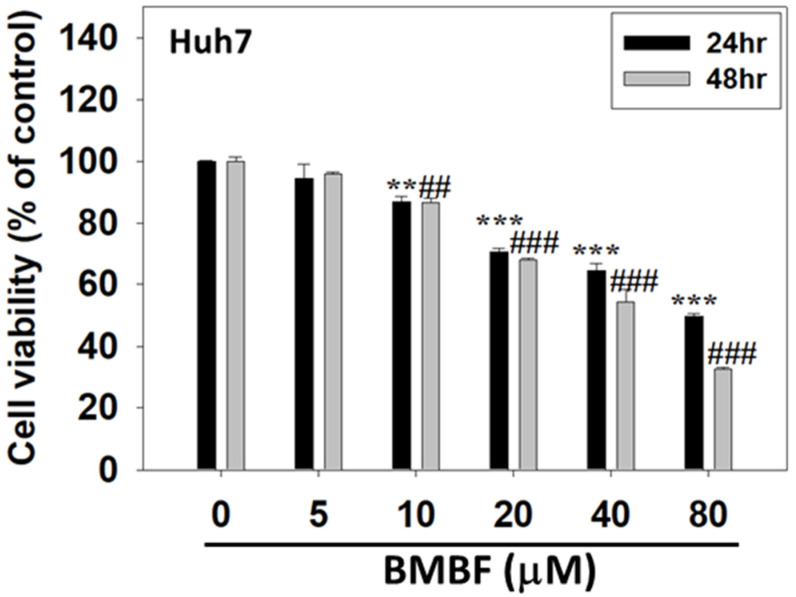
Anti-proliferation effect of BMBF in the Huh7 cells: After treatment with various concentrations of BMBF for 24 h and 48 h, the viable cells were determined using the CCK-8 kit. After treatment with the kit reagents, the optical density was measured at 450 nm using an ELISA multi-well plate reader. Data are represented as means ± SD (n = 3). The asterisks indicate statistical changes (** *p* < 0.01, *** *p* < 0.001, compared to the 24 h control; ## *p* < 0.01, ### *p* < 0.001, compared to the 48 h control).

**Figure 3 biomedicines-11-02027-f003:**
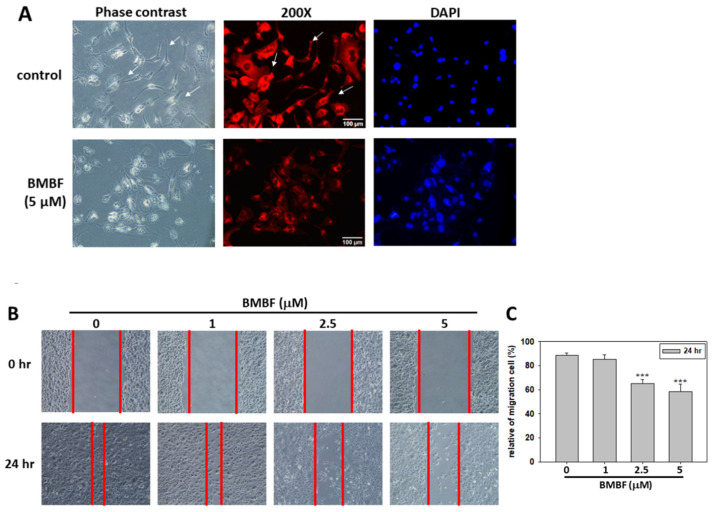
Effects of BMBF on the cytoskeleton and motility of Huh7 cells: (**A**) After treatment with or without BMBF (5 μM) for 48 h, the cytoskeleton of Huh7 cells was stained with TRITC-conjugated phalloidin, and the nuclei were stained with DAPI. The microscope image was taken (400×). White arrows pointed out the stained F-actin cytoskeleton. (**B**) The cell was scratched with a yellow pipette tip and photographed by a phase-contrast microscope under 100× magnification (0 h). Subsequently, the Huh7 cells were treated with BMBF for 24 h, and then they were observed and photographed (24 h). (**C**) The area of the cells that migrated into the scratched area was determined in 5 randomly selected fields by digital planimetry using ImageJ software. The area of cell migration was expressed as a percentage of the initial area (0 h). Data are represented as the means ± SD of three independent experiments (*** *p* < 0.001).

**Figure 4 biomedicines-11-02027-f004:**
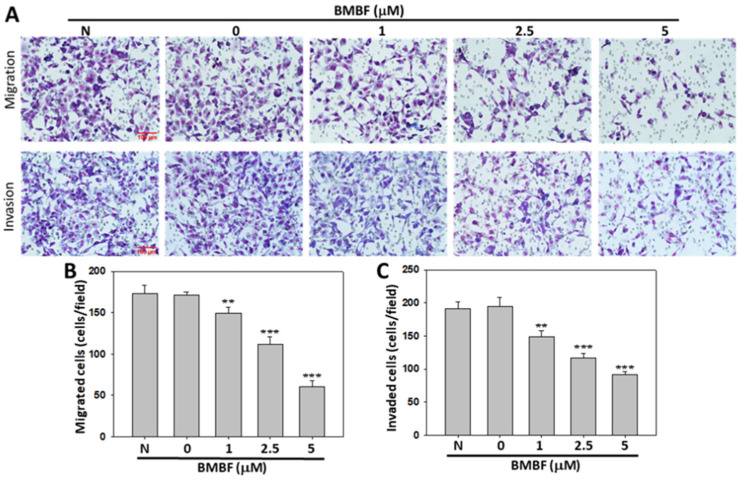
Inhibitory effects of BMBF on the migration and invasion of Huh7 cells: Huh7 cells (6 × 10^4^) were seeded onto the upper chamber, consisting of 8 μm pore-size filters coated without (upper panel) and with a Matrigel matrix, and the complete growth medium was placed in the lower chamber. After incubation for 24 h with or without BMBF, the filters were fixed for 10 min with methanol and stained with Giemsa for 1 h. The cells that had migrated or invaded into the lower surface of the filter were observed (200×) under microscopy and photographed (**A**) and counted in 5 randomly selected fields (**B, migrated cells; C, invaded cells**). Data are represented as the means ± SD of three independent experiments (** *p* < 0.01; *** *p* < 0.001). Scale bar = 100 μm.

**Figure 5 biomedicines-11-02027-f005:**
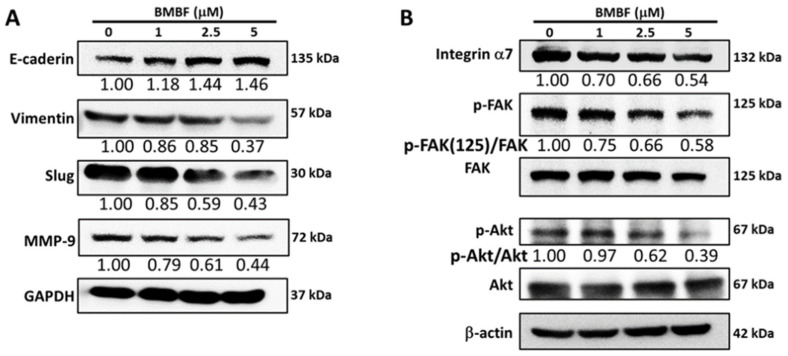
Effects of BMBF on the expression of EMT-related proteins, MMP9, integrin α7, and its downstream signal proteins: After treatment with BMBF for 24 h, the total cell lysates were prepared and subjected to Western blot analysis against specific antibodies, as indicated in the figure, (**A**), EMT-related proteins; (**B**), Integrin α7 and its downstream signal mediators. GADPH or β-actin was used as the loading control. The relative image density of each image was quantified by densitometry.

**Figure 6 biomedicines-11-02027-f006:**
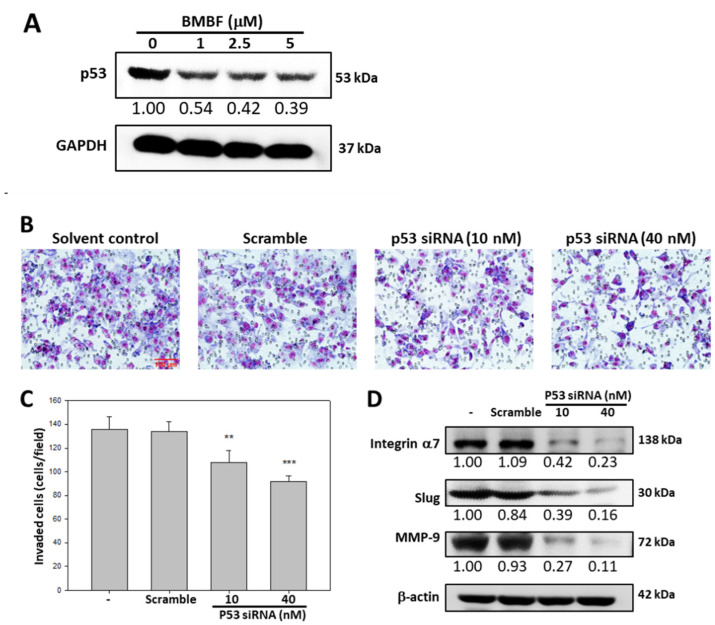
Anti-invasion effect of BMBF associated with downregulation of p53-mediated suppression of the expression of integrin α7, Slug, and MMP-9: (**A**) Effect of BMBF on the protein expression of p53 in Huh7 cells, evaluated by immunoblotting analysis. (**B**) Transfection of p53 siRNA affecting the invasion of Huh7 cells by Boyden chamber assay. After transfection with p53 siRNA for 48 h, the cells were seeded onto the upper chamber, consisting of an 8 μm pore-size filter coated with a Matrigel matrix, and then the complete growth medium was placed in the lower chamber and incubated for 24 h. The cells that invaded into the lower surface of the filter were observed (200×) under microscopy and photographed and counted in 5 randomly selected fields. Scale bar = 100 μm. (**C**) Data are represented as the means ± SD of three independent experiments (** *p* < 0.01, *** *p* < 0.001). (**D**) After transfection with p53 siRNA for 48 h, the total cell lysate was prepared, and then the expression of integrin α7, Slug, and MMP-9 was evaluated by immunoblotting analysis. β-actin was used as the loading control. The relative density of the images was quantified by densitometry.

**Figure 7 biomedicines-11-02027-f007:**
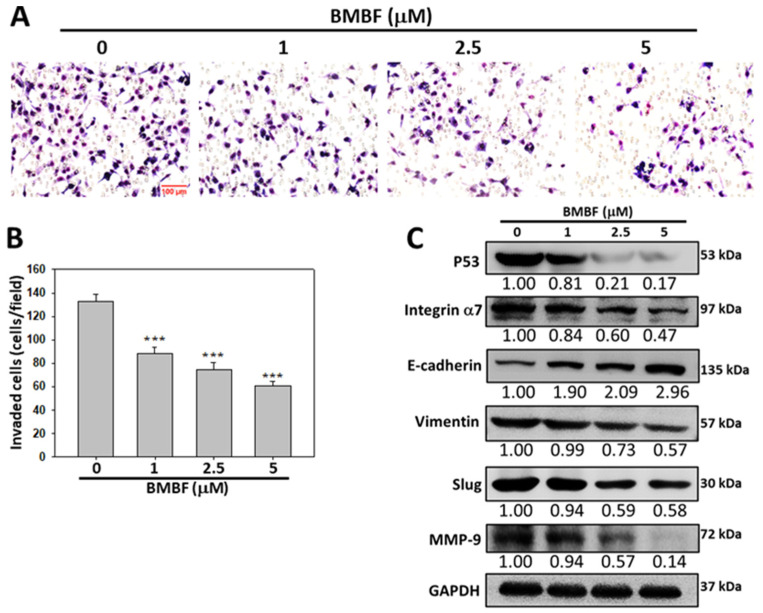
Inhibitory effect of BMBF on cell invasion of PLC/PRF/5 cells: (**A**) PLC/PRF/5 cells (2 × 10^5^) were seeded onto an upper chamber consisting of an 8 μm pore-size filter coated with a Matrigel matrix, and the complete growth medium was placed in the lower chamber. After incubation with BMBF for 24 h, the filters were fixed for 10 min with methanol and stained with Giemsa for 1 h. The cells that had invaded into the lower surface of the filter were observed (200×) under microscopy and photographed and counted in 5 randomly selected fields. (**B**) Data are represented as the means ± SD of three independent experiments (*** *p* < 0.001). Scale bar = 100 μm. (**C**) After treatment with BMBF for 24 h, the total cell lysates were prepared and subjected to immunoblotting analysis against specific antibodies, as indicated in the figure. GADPH was used as the loading control. The relative image density was quantified by densitometry.

**Figure 8 biomedicines-11-02027-f008:**
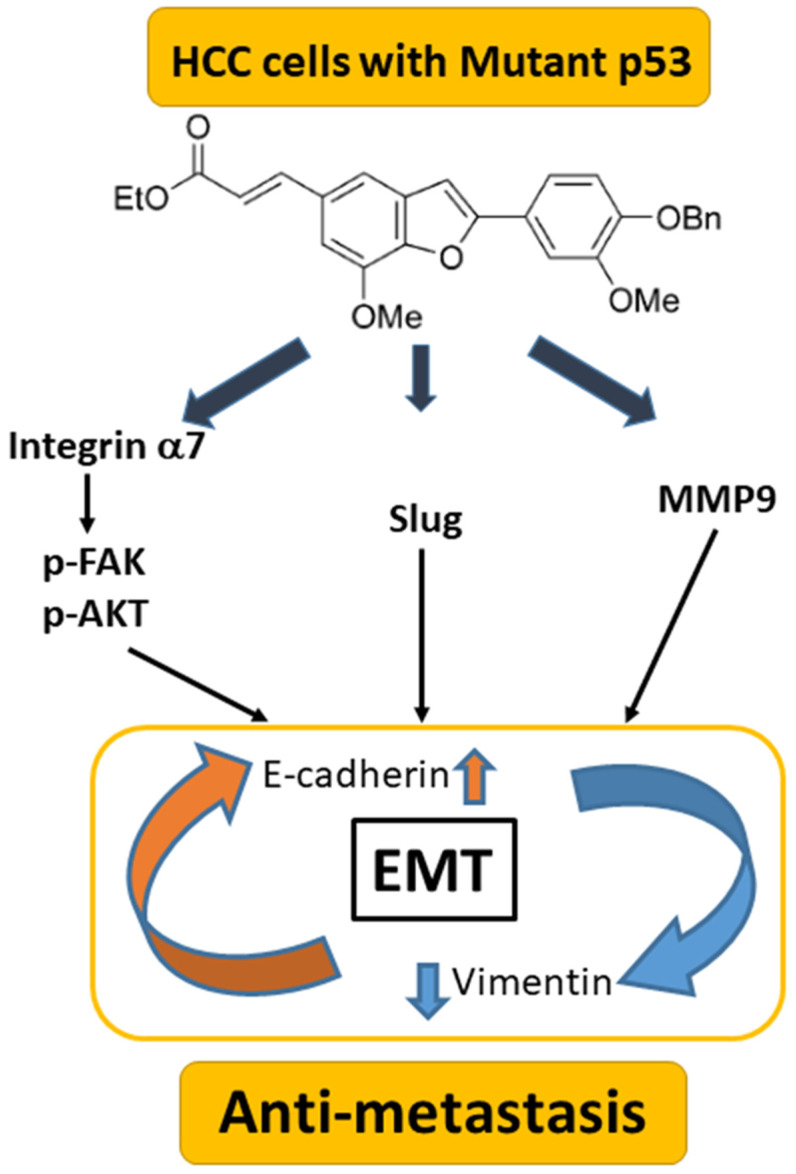
Summary of BMBF in HCCs: BMBF possesses anti-metastatic potential involving downregulated mutant p53 mediating alterations of integrin α7, EMT, and MMP9 in HCCs.

## Data Availability

All relevant data are within the paper.

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
