# Peer review of "2-(4-Benzyloxy-3-methoxyphenyl)-5-(carbethoxyethylene)-7-methoxy-benzofuran, a Benzofuran Derivative, Suppresses Metastasis Effects in P53-Mutant Hepatocellular Carcinoma Cells"

_biomedicines, 2023, doi:10.3390/biomedicines11072027_

Round 1

Reviewer 1 Report

The authors studied the pharmacological effect of benzofuran derivative (BMBF) in hepatocellular carcinoma (HCC) cells with p53 mutation. The authors described that BMBF inhibited the proliferation, migration and invasion of Huh-7 cells. The decreased expression of p53 was detected in the Huh-7 cells treated with BMBF. Induction of mesenchymal-epithelial transition was suggested to be induced in the Huh-7 cells treated with BMBF. The authors indicated that mutations in p53 involve in tumor growth and the regulation of metastasis. The reviewer agrees with the effect of the anti-proliferation and anti-migration. On the other hand, the reviewer can not understand the reason why the authors focused on the pharmaceutical effect of BMBF only in the HCC cells with p53 mutation. There is no evidences that indicate the relationship between the BMBF and p53 mutation. In addition, this relationship is still unclear because the present study did not contain the analyses of BMBF using HCC cells without p53 mutation. Therefore, the reviewer considers that there is no reason why the present study investigate the pharmaceutical effect of BMBF only in HCC cells with p53 mutation. If there is any evidence describing the relationship between the p53 mutation and the benzofuran derivative (or similar kinds of compounds), the authors should cite them and describe some comments in the introduction. Or the authors should analyze the pharmaceutical effect of BMBF not only in HCC cells with p53 mutation but also those with wildtype p53. In the reviewer’s opinion, the present manuscript (or study) is difficult to consider its significance.

The present manuscript does not have significant problem in the quality of English language.

Author Response

We thank the constructive comments given by Reviewer 1. The Review Report has been attached.

Reviewer 2 Report

In this manuscript, the authors demonstrated that noncytotoxic concentrations of BMBF exhibited antimigration and anti-invasion effects, and downregulated MTP53 levels in HCC cells with mutated p53 genes. This is an interesting study but it still needs revisions to be accepted for Biomedicines journal.

No.1: In this study, the authors showed BMBF downregulated the expression levels of MTP53 in Huh7 and PLC/PRF/5 cells. However, its underlying mechanisms were not discussed. This is thought to be a key point of this study, so that the author had better discuss the mechanisms more.

No.2: In “Introduction” section, the authors described that MTP53 promotes epithelial-to-mesenchymal transition (EMT), whereas WTP53 prevents EMT (Page2, line 8-9). The authors had better describe more about molecular mechanisms by which MTP53 promotes EMT by referencing previous reports. This is helpful for readers’ understanding of background.

No.3: The authors used the noncytotoxic concentrations of BMBF (5uM) in this study. This concentration is noncytotoxic for Huh7 cells; however, the safety for the normal cells (e.g., normal hepatocyte) was not evaluated. This should be conducted in this study.

No.4: There are some grammatical errors and typos. The authors had better ask an English proofreading to a native speaker of English. 

There are some grammatical errors and typos. The authors had better ask an English proofreading to a native speaker of English. 

Author Response

We thank your for agreeing with our current study. The 

Review Report has been attached.

Round 2

Reviewer 1 Report

The reviewer has understood the authors' responses. 

Although the reviewer considers the quality of English became fine, the final check should be performed.

Reviewer 2 Report

The authors adequately revised their manuscript according to the reviewers’ comments. Therefore, this manuscript seems acceptable for the publication in the journal of “Biomedicine”.